# Real-World Goal Setting and Use of Outcome Measures According to the International Classification of Functioning, Disability and Health: A European Survey of Physical Therapy Practice in Multiple Sclerosis

**DOI:** 10.3390/ijerph17134774

**Published:** 2020-07-02

**Authors:** Kamila Řasová, Patrícia Martinková, Bernadita Soler, Jenny Freeman, Davide Cattaneo, Johanna Jonsdottir, Tori Smedal, Anders Romberg, Thomas Henze, Carme Santoyo-Medina, Peter Feys

**Affiliations:** 1Department of Rehabilitation, Third Faculty of Medicine, Charles University, Prague 108 00, Czech Republic; 2Department of Statistical Modelling, Institute of Computer Science of the Czech Academy of Sciences, Prague 182 07, Czech Republic; martinkova@cs.cas.cz; 3Neurology, Pontificia Universidad Católica de Chile, Santiago 3580000, Chile; bernarditasoler@gmail.com; 4Neurology, Hospital Doctor Sótero del Rio, Santiago 8320000, Chile; 5REVAL, Rehabilitation Research Center, Faculty of Rehabilitation Sciences, Hasselt University, Hasselt 3500, Belgium; peter.feys@uhasselt.be; 6Faculty of Health, University of Plymouth, Devon PL6 8BH, UK; jenny.freeman@plymouth.ac.uk; 7IRCCS Fondazione Don Carlo Gnocchi, Larice Lab, 20148 Milan, Italy; dcattaneo@dongnocchi.it (D.C.); jjonsdottir@dongnocchi.it (J.J.); 8Norwegian Multiple Sclerosis Competence Centre, Department of Neurology and Department of Physiotherapy, Haukeland University Hospital, 5021 Bergen, Norway; tori.smedal@helse-bergen.no; 9Physiotherapy, Masku Neurological Rehabilitation Centre, 21250 Masku, Finland; anders.romberg@neuroliitto.fi; 10Specialist Practice in Neurology, 93059 Regensburg, Germany; thomas.henze@outlook.com; 11Neurology-Neuroimmunology Department, Neurorehabilitation Unit, Multiple Sclerosis Centre of Catalonia (Cemcat), Vall d’Hebron University Hospital, 08035 Barcelona, Spain; csantoyo@cem-cat.org

**Keywords:** goals, measures, international classification of functioning, disability and health, physical therapy, multiple sclerosis, questionnaire survey, cluster analysis, Europe

## Abstract

Goal setting is a core component of physical therapy in multiple sclerosis (MS). It is unknown whether and to what extent goals are set at different levels of the International Classification of Functioning, Disability and Health (ICF), and whether, and to which, standardized outcome measures are used in real life for evaluation at the different ICF levels. Our aim was to describe the real-world use of goal setting and outcome measures in Europe. An online cross-sectional survey, completed by 212 physical therapists (PTs) specialized in MS from 26 European countries, was conducted. Differences between European regions and relationships between goals and assessments were analyzed. PTs regularly set goals, but did not always apply the Specific, Measurable, Achievable, Realistic, Timed (SMART) criteria. Regions did not differ in the range of activities assessed, but in goals set (e.g., Western and Northern regions set significantly more goals regarding leisure and work) and outcome measures used (e.g., the Berg Balance Scale was more frequently used in Northern regions). Quality of life was not routinely assessed, despite being viewed as an important therapy goal. Discrepancies existed both in goal setting and assessment across European regions. ICF assists in understanding these discrepancies and in guiding improved health-care for the future.

## 1. Introduction

Multiple sclerosis (MS) is a chronic, inflammatory demyelinating and neurodegenerative immune-mediated disease of the central nervous system that causes neurological signs and symptoms. Impaired functioning and disability are the core experience of MS patients; numerous symptoms and considerable disabilities negatively influence a range of functions and limit participation and quality of life [1]. Rehabilitation is recommended in addition to pharmacological treatment to optimize daily functional activities, mobility, occupation, communication, social integration, and quality of life [2,3].

Physical therapy (PT) primarily treats physical functions with the aim of promoting functional independence, preventing complications, and of enhancing overall quality of life. To achieve this, PT uses a variety of techniques and methods that can be broadly divided into four categories: physical activity (fitness/endurance/resistance) training, motor/skill acquisition (individualized therapy-led intervention), neuro-proprioceptive “facilitation, inhibition”, and technology-based PT [4]. PT interventions can lead to improvements in all categories defined by the International Classification of Functioning, Disability and Health (ICF) [5,6], which have been documented by clinically meaningful patient-reported outcomes [7,8,9].

The World Health Organization (WHO) suggests that the key indicator of the health of populations is functioning [10]. In this context, the ICF is an integrative and biopsychosocial approach that comprehensively describes the impact of a health condition on an individual’s functioning. As a universal framework, the ICF could be used and applied across countries, health conditions, and healthcare settings. Within its conceptual framework, functioning and disability are associated with the interaction of different components, namely the individual’s body functions and body structures, activities and participation, environmental factors, and personal factors [1]. In MS, an ICF Core Set has been defined, designed to enable a brief description of functioning in persons with MS in multidisciplinary assessments in both clinical and research settings [5,11].

However, functioning and health are not only an outcome but also the starting point in the assessment of a patient. Rehabilitation is an essential care needed when a person is likely to experience limitations. Evidence and availability of standardized outcomes measures are growing in MS rehabilitation as demonstrated by the publication of both systematic and narrative reviews in this area [12,13,14,15]. However, evidence concerning its real-world application is lacking. 

The WHO has included rehabilitation as part of the achievement of universal health coverage in the General Programme of Work 2019–2023 as one of its strategic priorities for meeting Sustainable Development Goal 3: “Ensure healthy lives and promote well-being for all at all ages”. This means that all people should receive quality health services that meet their needs without being exposed to financial hardship in paying for the services. 

While the need for rehabilitation is increasing in many parts of the world, health systems do not have the capacity to provide the necessary rehabilitation services [16] to effectively integrate rehabilitation into national health plans and budgets and achieve the goal of universal health coverage. Actions required include: strong leadership and governance; adequate funding for rehabilitation; efficient service delivery models; a multidisciplinary rehabilitation workforce with standardized outcome measures; affordable assistive products; and integration of rehabilitation data into a health information system [16].

Today only a few countries like France, the United Kingdom (UK), and Sweden have a system of monitoring the quality of physiotherapy (PT) in MS [12]. This is facilitated by organizations such as the National Institute for Health and Care Excellence (UK) who produce national guidelines, and professional associations such as Rehabilitation in Multiple Sclerosis (RIMS) who document current practice and identify areas for improvement within the rehabilitation field [3].

This study aims to describe the real-world use of goal setting (in which ICF domains are set as a goal), assessment (in which ICF domains are assessed), and outcome measures used (which of the outcome measures for a given ICF domain are used) in MS physical therapy. It also aims to evaluate whether there are differences across European regions and to point to existing disparities in assessment outcomes and targets in PT in MS across Europe.

## 2. Materials and Methods 

A descriptive, cross-sectional survey of physical therapists (PTs) who work with MS patients using convenience sampling was undertaken. The survey was part of the COPHYREQUEST project, of which other results are reported elsewhere [4,17,18,19].

This study focuses on survey items pertaining to goal setting, the functions assessed, and the outcome measures used. Respondents (PTs) were asked to rate, on a five-point Likert scale, whether and how often (never, rarely, sometimes, often, and almost always) each body function and structure, activity and participation level, were goals of their therapy in MS patients. They were also asked about the number of goals they set, and their utilization of the SMART (Specific, Measurable, Achievable, Realistic, Timed) approach [20] to setting and monitoring goals. Questions regarding which domains (on body function, activity, and participation level) were assessed (and how often) were also included in the survey, and these were further detailed by questions on the outcome measures being used. Using the ICF browser, each domain was labelled with its respective ICF code according to body function, activity, and participation level (Appendix A). 

European regions were defined according to the United Nations Statistics Department [21]. To test differences between regions, we employed the Pearson Chi-square test with a simulated *p*-value [18], due to very small counts in some cells. To compare the use of goals and assessment in the same ICF domain, we used paired t-tests. To account for multiple comparisons, the Benjamini-Hochberg correction was applied [22]. Results were reported as statistically significant if the corrected *p*-value was lower than 0.05. 

Clusters of correlated assessment domains (body functions and activities that are jointly assessed) were identified using a hierarchical clustering approach with the distance based on polychoric correlations and Ward’s linkage [23]. Statistical environment R, version 3.6.0, and its libraries were used throughout the analyses. 

## 3. Results

### 3.1. Respondents

A total of 212 respondents from 115 workplaces across 26 European countries participated in the study (response rate 53%). Of the total respondents, 16.5% were from Eastern Europe (Czech Republic, Poland, Romania, and Slovakia), 9.9% from Western Europe (Austria, Belgium, France, Germany, Netherlands, and Switzerland), 42.9% from Southern Europe (Croatia, Former Yugoslav Republic of Macedonia, Greece, Italy, Portugal, Serbia, Slovenia, Spain, and Turkey), and 30.6% from Northern Europe (Denmark, Estonia, Finland, Ireland, Norway, Sweden, and United Kingdom) [4]. 

### 3.2. Goal Setting 

Ninety-four percent of respondents specified that goal setting was considered an integral part of PT. It was reported that in 99% of cases the patient was involved in the goal setting process, with family involvement in 67% of cases. Eighty-three percent of respondents reported that this goal setting process involved more than one rehabilitation professional. 

Sixty-one percent of respondents reported using SMART goals significantly more often in the Northern region. Sixteen percent reported using Goal Attainment Scaling (GAS), and 39% using other scales to measure goal achievement. Most of the respondents (81%) typically set 1–3 goals with their patients, 14% usually set 4–5 goals, with 5% reporting that they set more than 5 goals. There were no statistically significant differences between European regions in this aspect. 

The body functions cited most frequently as a goal of therapy were trunk muscle power and tone (set as a goal at least “sometimes” by 96% of respondents). The least-evaluated body function goal was genital-reproductive functions (set as a goal at least “sometimes” by only 19%, never set as a goal by 50% of respondents). The most frequent activities listed as goals of therapy were walking and moving around using equipment, and changing and maintaining body position (see Figure 1).

Between the European regions, significant differences were found in goal setting of the visual or oculomotor functions, which were more often used as a goal of therapy in the Eastern and Southern European regions; and in leisure, and work, which were more often used as goals in the Western and Northern European regions. 

### 3.3. Assessed International Classification of Functioning, Disability and Health (ICF) Domains 

The most frequently assessed body functions were exercise tolerance, gait pattern function, and trunk muscle power and tone. The least-evaluated were genital-reproductive functions. The most frequent activity domains evaluated were walking and moving around using equipment, and hand and arm use (see Figure 2). There were no significant differences between regions in domains assessed.

A similar pattern to goal setting can be found in the use of assessment domains (Figure 1 vs. Figure 2), but there were some significant differences (Table 1). The biggest discrepancy between goal setting and assessment was found for quality of life, which is much more often set as a goal than it is actually assessed by the PTs.

### 3.4. Outcome Measures Used for Assessment

Figure 3 shows how often the PTs used various outcome measures for assessment of the body functions and structures, Figure 4 depicts the use of outcome measures for assessment of activities/participation.

The Modified Ashworth scale for assessment of muscle tone function (used at least “sometimes” by 68% of respondents), spatio-temporal parameters for gait pattern (61%), and the Berg Balance Scale (59%) were the most commonly used outcome measures at body function level. The Timed Up and Go test (55%), the 6-min walking test (52%) and the 10-Meter Walk Test at normal speed (44%) were the most commonly used outcome measures at activity level, all of them in the walking and moving domain.

While the four European regions did not differ in the domains assessed, they differed in the specific outcome measures they used. Significant differences were found in number of outcome measures including the Berg Balance Scale, the Dynamic Gait Index, the Fatigue Severity Scale, the 10-Meter Walk Test at normal and maximal speed, the 12-item-MS Walking Scale, the Nine Hole Peg Test and the MS Impact Scale (mainly used in Northern European region), the Tinetti mobility test, the Trunk Control Test, the Hauser Ambulation Index and Incapacity Status Scales (mainly used in Southern and Western European regions), the Motricity Index, the Medical Research Council Scale, the 2-min Walk Test, the Functional Ambulation Categories and Action Research Arm Test (mainly used in Western European regions), the 6-min Walk Test (mainly used in Northern and Western European regions), and Functional Assessment of MS (mainly used in Southern and Northern European regions). It appears that while there are no significant differences in the domains assessed, when it comes to specific outcome measures used, these are less often used in Eastern European region (Figure 5, Appendix A). 

### 3.5. Relationships Between Domains Assessed

We found three clusters of body functions and activities usually assessed jointly (Figure 6), namely: 1. Mobility; 2. Self-care; and 3. Body functions (other than motor functions, e.g., urinary function). It seems that body functions and activities from the third cluster were those that were significantly less frequently assessed than they should have been according to their setting as a goal. 

When the differences between European regions were explored, we found that the Northern and Southern regions assessed the three clusters simultaneously in contrast with the Eastern region where the third cluster was not measured, and the Western region where the strength of the relationship between these clusters was less clear (see Figure 7 vs. Figure 6). 

## 4. Discussion

PT is a problem-solving and educational process in which it is recommended that goal setting is a core component of the process [24,25]. Our results showed that almost all PTs used goal setting, typically setting 1–3 goals. Different approaches to goal setting were used, and whilst over half (61%) of PTs used SMART criteria to guide them, only 16% used the GAS. It is therefore apparent that there is considerable potential for PTs to use goal setting as a method of enhancing their effectiveness. For example, the GAS has been shown to be a useful therapeutic tool for evaluating goal achievement, providing a mechanism for communicating and monitoring treatment progress between patients and therapists [26]. 

The questionnaire addressed all levels of the ICF model [25], which enhances our understanding into which domains were addressed by PTs in their clinical practice. It allowed an international perspective to compare the different outcomes between countries. Although the ICF core set in MS [6] indicates which areas of functioning should be measured, it does not recommend how this should be achieved. Furthermore, the ICF browser does not categorize all clinical signs into a single body function or structure such as ataxia.

Our results are in line with recommendations for practice [27]. Typically, PTs mainly aim at influencing functions such as exercise tolerance, muscle power and tone, energy level, and pain. Except for the Eastern and Southern European regions, other functions such as seeing, oculomotor, bladder, and bowel control were not indicated as a PT goal. 

Collaborative goal-setting with patients is recommended [24,25,26]. This survey indicated that walking and moving, changing and maintaining body position, using arms and hands, and self-care were the most common activities defined as a goal across all regions. In the Eastern region, the goals to improve self-care and major life areas were chosen less frequently. In the Eastern region, interdisciplinary teams were less frequently used [19], and hence it is possible that PTs considered that other professions, such as occupational therapists, covered this domain.

Many different measures were used to address different levels of the ICF [28]. In our questionnaire, PTs first chose from the listed domains (body functions and activities), and subsequently were asked to specify which of the different outcome measures they used for each of the assessed body functions or activities. Whilst almost all domains were assessed, only a few outcome measures were used. In line with other studies [29,30], we confirmed statistically significant regional differences in the use of many outcome measures. We tried to find some patterns in the PT’s choice between competing instruments in each domain, but it seems (in line with Haigh et al.’s 2001 study [29]) that the use of outcome measures is related to specific regional contexts rather than to the psychometric properties or appropriateness of the instrument. One apparent pattern is in the less-frequent use of specific outcome measures, such as those related to walking, pain/fatigability, energy levels, and major life areas in Eastern European region.

MS is an individually variable and unpredictable disease, which needs evaluation at different assessment levels (impairment, disability, handicap, quality of life), including patient-reported outcome measures. Our team, the Special Interest Group in Mobility of RIMS, has a long-term interest in understanding what is the most valuable assessment approach to address all levels of patient/therapist perspectives. Firstly, in the narrative review [2], drawbacks in current practice and research were defined, one of them being the non-systematic use of many different measures, making it difficult to directly compare the effectiveness of interventions. Secondly, several teams of specialists have tried to develop a core set of clinical functional assessments for MS rehabilitation [31,32], others are working on gaining a consensus to harmonize both assessments and interventions [33], but still without clear recommendations. Resources, such as the Rehabilitation Measures Database (https://www.sralab.org/rehabilitation-measures), which details operational instructions and the psychometric properties of many outcome measures, is helpful for facilitating the use of outcome measures in clinical practice and for designing research. Thirdly, a prospective multicenter trial was realized with the aim of identifying the most relevant measurement of mobility for use in physical rehabilitation of people with MS (pwMS) [34]. Responsiveness, clinically meaningful improvement, and real changes of measures, demonstrate great heterogeneity. For the mildly disabled pwMS, the Multiple Sclerosis Walking Scale–12 and the physical subscale of the Multiple Sclerosis Impact Scale–29 were the most sensitive measures in detecting improvements in functional mobility after physical rehabilitation, while for the moderate to severely disabled pwMS, the Rivermead Mobility Index and the capacity test 5-Repetition Sit-to-Stand Test seemed to be useful [34].

Two decades ago, disability and functional independence were the predominant domains measured in neurological patients [29,30]. Our survey indicates that nowadays, a broad range of body function and activity domain measures were utilized by most (80%) of the respondents. This may be reflecting a shift in clinical practice towards a more systematic use of the ICF model. This model offers very good tools for assessment (ICF Assessment Sheet), goal setting (ICF Categorical Profile), for documentation of interdisciplinary treatment processes (ICF Intervention Table), and for evaluation as to whether goals are addressed (ICF Evaluation Display), see (https://www.icf-casestudies.org). We are convinced that this person-centered concept has significant potential for the future, because it reflects the components of individual functioning together with the contextual components [10]. 

Although our results are unsurprising in demonstrating that PTs mainly assess body functions connected with mobility, notable is the frequency of assessment of visual and oculomotor functions. Concerning the assessment of fatigue and pain, the Visual Analogue Scale is the outcome measure most commonly used. Of note, although these symptoms were commonly identified as important goals in treatment (more than 75%), fewer respondents (50%) used outcome measures to examine them. In contrast, most respondents considered muscle tone to be a frequent aim of therapy and used standardized body function measures to assess this. Of interest, the scale used to evaluate muscle tone by most respondents was the Ashworth Scale, despite it being widely criticized as having significant limitations in terms of its clinical value and psychometric properties [30].

This survey demonstrates that physical fitness (heart rate and Borg Rating of Perceived Exertion) and walking (6-min walking test and spatio-temporal parameters) are commonly included in MS clinical practice. One could postulate that the ICF conceptual model has been influential in encouraging therapists to assess a breadth of domains when examining their rehabilitation practice. Furthermore, the exponential increase in evidence demonstrating the benefits of aerobic training in MS [5,7] means that assessments relevant to the evaluation of this intervention are now often incorporated within clinical practice, namely measures of heart rate and perceived exertion. The assessment of gait remains common, unsurprisingly given the abundant evidence, which demonstrates gait dysfunction as a key symptom, which is of importance to people with MS, and which is evident from the very early stages of the disease [35]. 

Although respondents acknowledged that they often set quality of life as the goal of their therapy, they do not assess it so frequently in their MS patients. Thompson (2000) [36], amongst others, has emphasized that outcome measures should attempt to capture the entire impact of the rehabilitation process, looking not just at activities and participations, but also on quality of life, coping skills and self-efficacy. Our survey highlights that almost twenty years later, this appears to remain lacking. 

## 5. Conclusions

This research showed that in most of Europe, there remains the need to use a common framework for establishing and evaluating treatment goals and interventions in everyday practice. The ICF model provides such a framework, and there are numerous standardized and scientifically validated outcome measures available for use. In this context, it is important to develop consensus in order to optimize PT.

## Figures and Tables

**Figure 1 ijerph-17-04774-f001:**
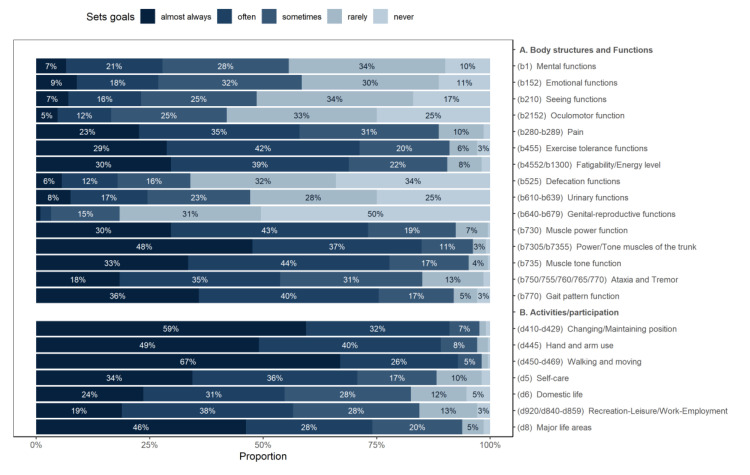
Body structures and functions and activities/participation used as goals in therapy by physical therapists (PTs) in multiple sclerosis (MS), from an International Classification of Functioning, Disability and Health (ICF) perspective.

**Figure 2 ijerph-17-04774-f002:**
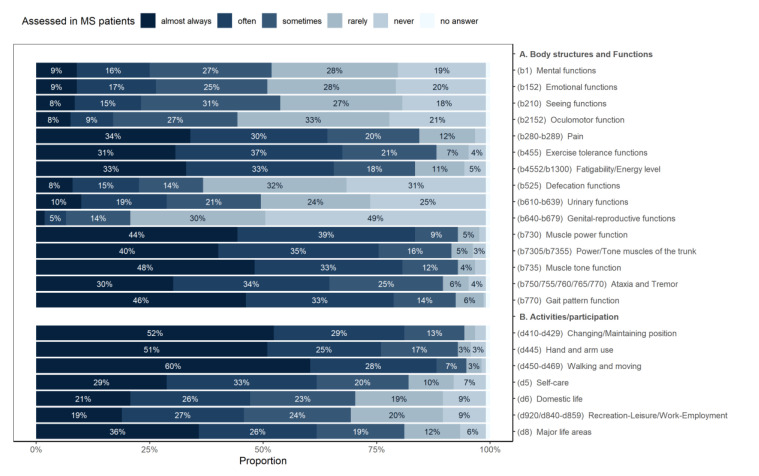
Body structures and functions and activities/participation assessed in MS patients, from an ICF perspective.

**Figure 3 ijerph-17-04774-f003:**
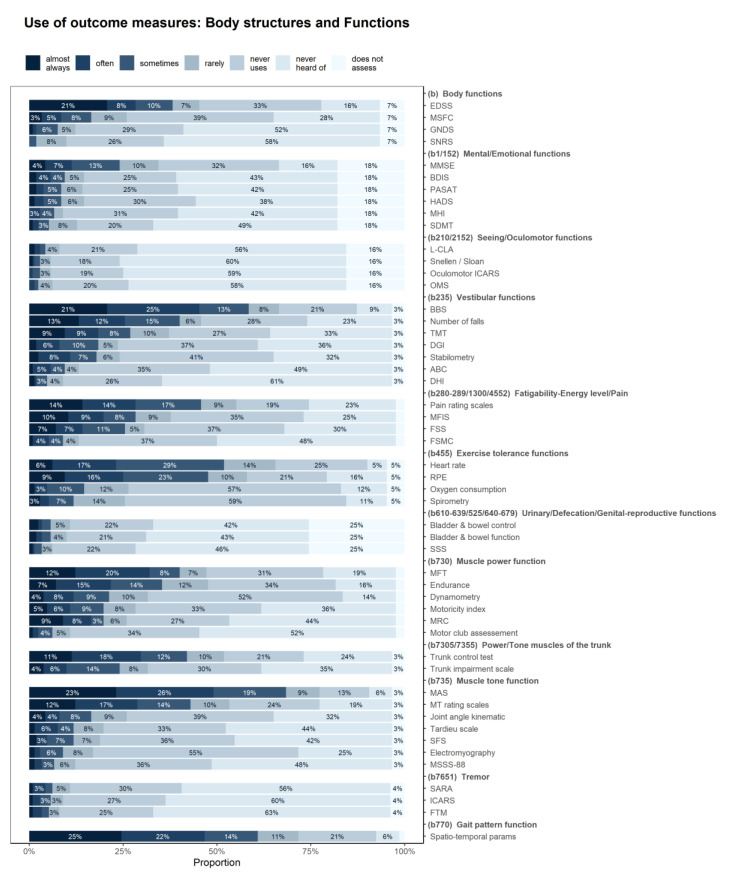
Frequency of use of the different outcome measures for the assessment of the different body functions and structures in MS patients. For abbreviations, see Appendix A.

**Figure 4 ijerph-17-04774-f004:**
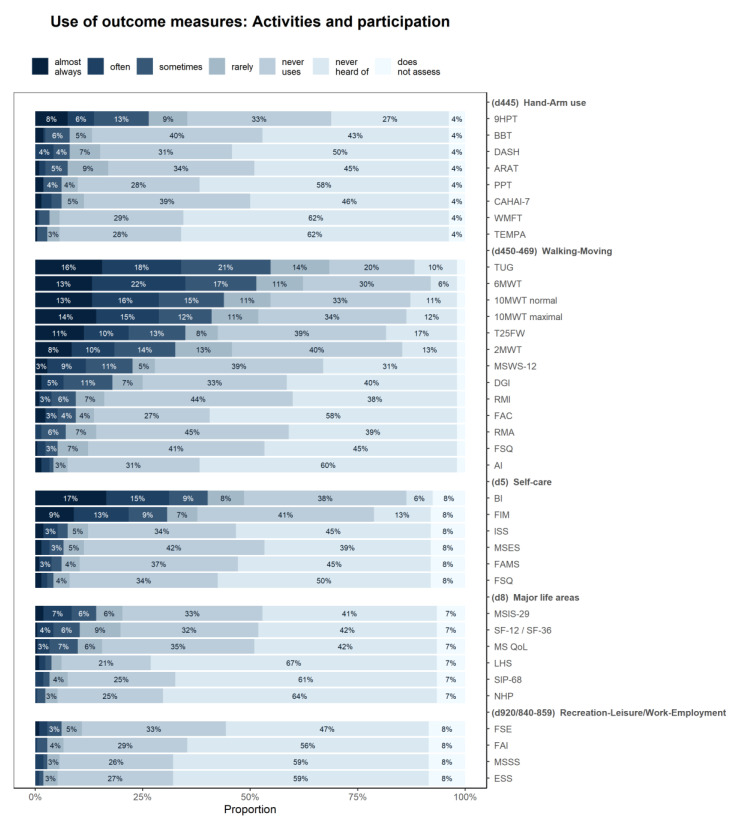
Frequency of use of the different outcome measures for the assessment of the different activity and participation levels in MS patients. For abbreviations, see Appendix A.

**Figure 5 ijerph-17-04774-f005:**
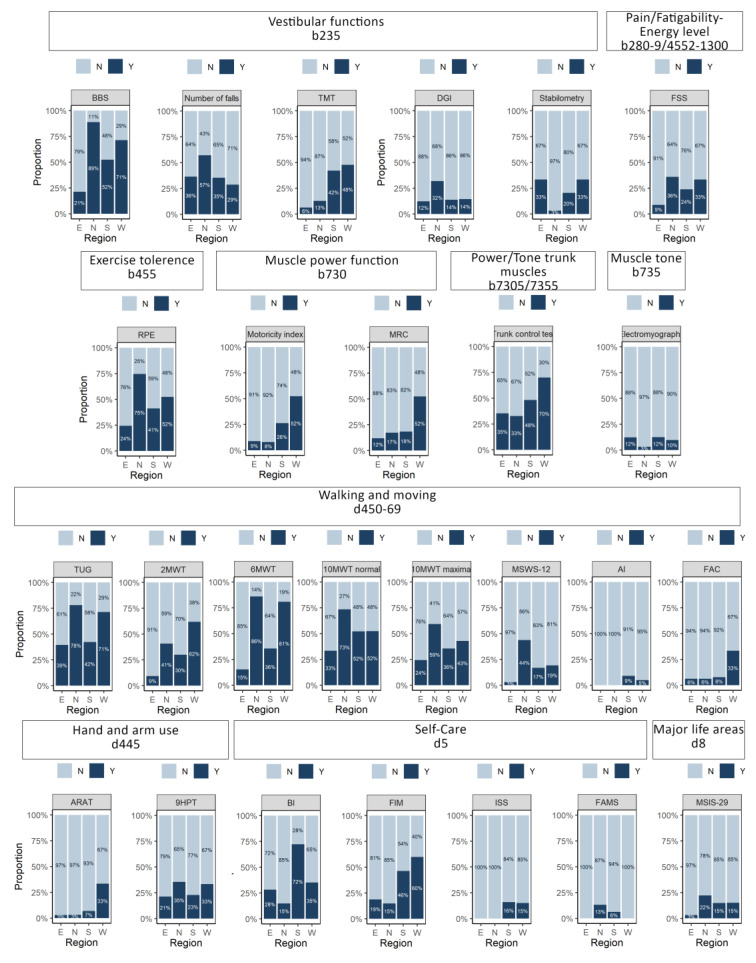
Significant differences between European regions in the use of outcome measures for body structure/function and activity/participation level. BBS: Berg Balance Scale, TMT: Tinetti Mobility Test, DGI: Dynamic Gait Index, FSS: Fatigue Severity Scale, RPE: Borg Rating of Perceived Exertion, MRC: Medical Research Council Scale, TUG: Timed Up and Go, 2MWT: 2-Minute Walking Test, 6MWT: 6-Minute Walking Test, 10MWT normal: 10-Meter Walk Test at normal speed, 10MWT maximal: 10-Meter Walk Test at maximal speed, MSWS-12: 12-Item Multiple Sclerosis Walking Scale, AI: Hauser Ambulation Index, FAC: Functional Ambulation Category, ARAT: Action Research Arm Test, 9HPT: Nine Hole Peg Test, BI: Barthel Index, FIM: Functional Independence Measure, ISS: The Incapacity Status Scale, FAMS: Functional Assessment of Multiple Sclerosis, MSIS29: Multiple Sclerosis Impact Scale. Also see Appendix A.

**Figure 6 ijerph-17-04774-f006:**
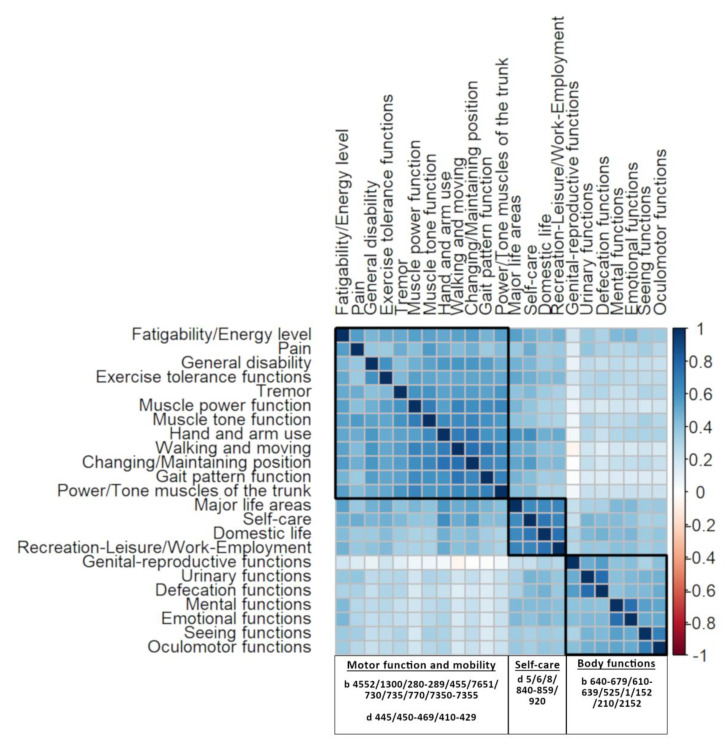
Relationships between individual body functions/activities assessed in all centers.

**Figure 7 ijerph-17-04774-f007:**
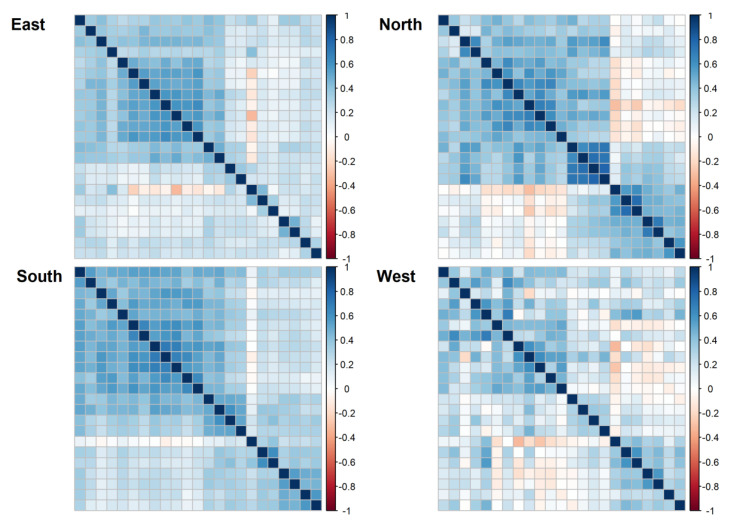
Relationships between individual body functions/activities assessed in four European regions. Ordering of body functions/activities kept the same as in Figure 6.

**Table 1 ijerph-17-04774-t001:** Use of different body structures and functions and activities/participation as goals of the therapy (Goals) and use of assessments to measure these body functions and activities (Assess), both measured on scale of 1: never; 2: rarely; 3: sometimes; 4: often; and 5: almost always.

Body Function or Activity	Goals	Assess	Difference		d	T	*p*
	Mean	Mean	Mean	SD			
Mobility (walking, wheelchair handling)	4.58	4.45	0.12	0.80	0.15	2.23	0.057
Changing and maintaining body position	4.47	4.28	0.20	0.99	0.20	2.86	0.015*
Using arms and hands	4.35	4.19	0.16	1.08	0.15	2.17	0.057
Trunk control	4.28	4.06	0.22	0.99	0.22	3.20	0.009*
Quality of life	4.13	3.75	0.38	1.17	0.33	4.74	<0.001*
Muscle tone	4.06	4.21	−0.15	1.01	−0.15	−2.19	0.057
Gait/wheelchair pattern functions	4.01	4.19	−0.18	1.07	−0.17	−2.45	0.037*
Muscle power function	3.95	4.21	−0.27	1.03	−0.27	−3.84	0.001*
Self-care	3.92	3.67	0.24	1.26	0.19	2.80	0.015*
Exercise tolerance/Physical fitness	3.88	3.84	0.04	1.18	0.04	0.53	0.600
Fatigue	3.87	3.79	0.08	1.08	0.08	1.08	0.325
Pain	3.68	3.82	−0.14	1.04	−0.13	−1.93	0.093
Leisure and work-related activities	3.57	3.26	0.31	1.14	0.27	3.87	0.001 *
Ataxia and tremor	3.56	3.82	−0.24	1.23	−0.20	−2.81	0.015 *
Domestic life	3.56	3.30	0.26	1.23	0.21	3.03	0.012 *
Psychological functions	2.83	2.67	0.15	1.20	0.13	1.85	0.104
Mental functions	2.80	2.67	0.12	1.22	0.10	1.47	0.186
Visual function	2.62	2.68	−0.06	1.12	−0.06	−0.80	0.446
Bladder control	2.54	2.63	−0.08	1.19	−0.07	−0.99	0.356
Oculomotor function	2.38	2.48	−0.11	0.98	−0.11	−1.55	0.170
Bowel control	2.24	2.37	−0.13	1.18	−0.11	−1.58	0.170
Sex functions	1.72	1.81	−0.08	0.94	−0.08	−1.18	0.292

Diff: difference between grade for goals (how often is a given domain set as a goal of therapy by the physiotherapist in their patients?) and assessments (how often does the physiotherapist assess given domain in their patients?); d: Cohen’s d; * Significant differences between goal and assessments (*p* < 0.05, displayed are *p* values of paired t tests, corrected for multiple comparisons).

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
