# Peer review of "Real-World Goal Setting and Use of Outcome Measures According to the International Classification of Functioning, Disability and Health: A European Survey of Physical Therapy Practice in Multiple Sclerosis"

_ijerph, 2020, doi:10.3390/ijerph17134774_

Round 1

Reviewer 1 Report

The article Real-World Goal Setting and Use of Outcome Measures According to the International Classification of Functioning Disability and Health 5 (ICF): A European Survey of Physical Therapy Practice in Multiple Sclerosis written by Kamila Rasova et al. raises a very interesting and important issue of real-world use of goal setting and outcome measures in the context of Multiple Sclerosis and their distribution in Europe. In the Reviewer's opinion, the material is selected carefully, line of thinking and analysis are performed properly, but this paper has several problems that the Authors must solve. First of all, the presentation of results (quality of presented graphics). Secondly, the article should be enriched by some discussions (opinions of the Authors) and not provide a dry report only.

Introduction

In the introduction section, the reader may feel a little bit unsatisfied about the characteristics and importance of the quality of physiotherapy and related activities in the context of multiple sclerosis itself. The problem should be formulated more precisely.

Materials

  • line 108, bracket error

Result

The basic and most important problem that the Authors of this article must solve is the very poor quality of presented graphics. Almost all of the presented figures are completely unreadable. 

The values in the charts are illegible, which was already a big problem in the reviewing of this work. I am asking for the Authors to print a paper version of the paper and try to guess without a magnifying glass what is presented on e.g Fig 4 (this seems to be one of the most interesting results of this work, but unfortunately invisible). The reviewer does not know if this is a problem at the editorial level of the article, but necessarily it must be solved.

Discussion

In the section results (lines 178-189) Authors mention several different indexes or scale types. In Discussion, the Authors characterize what kind of scale is using more frequently. Which of them in the Authors opinion bring the most valuable information? I understand that the purpose of the work was the- characterization, but the thematic scope of discussion could be enriched by the attempt of assessment of physical tests quality in the context of MS

Other comments:

The statistical analysis in the Reviewer's opinion is performed properly, but there is a lack of justification for using the parametric tests? 

Author Response

Prague, June 22, 2020

Dear Editors,

Thank you very much for the review of our manuscript entitled ““Real-world goal setting and use of outcome measures according to the International Classification of Functioning Disability and Health (ICF): A European survey of physical therapy practice in multiple sclerosis” by Řasová K., Martinková P., Soler B., Freeman J., Cattaneo D., JonsdottirJ., Smedal T., Romberg A., Henze T., Santoyo-Medina C. and Feys P.

We have revised the manuscript according to reviewers’ suggestions. Changes are highlighted throughout the text of the paper. Responses to each reviewer’s comment are also commented separately below.

We are glad that the reviewers’ view the paper as raising important questions, informative and well written. We believe that our revisions have further improved the manuscript and hope that it is now suitable for publication  in the International Journal of Environmental Research and Public Health.

Sincerely,

                                                           Assoc. prof. PhDr. Kamila Řasová, Ph.D.

The article Real-World Goal Setting and Use of Outcome Measures According to the International Classification of Functioning Disability and Health (ICF): A European Survey of Physical Therapy Practice in Multiple Sclerosis written by Kamila Rasova et al. raises a very interesting and important issue of real-world use of goal setting and outcome measures in the context of Multiple Sclerosis and their distribution in Europe. In the Reviewer's opinion, the material is selected carefully, line of thinking and analysis are performed properly

Thank you for your positive feedback.

… but this paper has several problems that the Authors must solve. First of all, the presentation of results (quality of presented graphics).

Thank you for this important point:

In order to make the figures readable, we have split original Figure 1A+B into two separate figures (1 and 2), and original Figure 2A+B into separate figures (3 and 4).

For original Figure 3 (now Figure 5) we have prepared Figures with better resolution (larger font), but these need to be combined into new Figure 5 (we are still working on this).  

For original Figure 4, we think we again need to split this into two separate tables (new Figure 6 and 7, see attached). However, I would need you to cut the white space around Figure 6, add the boxes with description of the clusters (we are still working on this).
Finally, we are still working on new Supplementary Table additional to Figure 3 with named the Domains. 

We are sorry, but we are still working on improvement of the graphs – it is technically more demanding than we expected. May we kindly ask you to postpone the term for graphs preparation?

Secondly, the article should be enriched by some discussions (opinions of the Authors) and not provide a dry report only.

We have enriched the article by some discussions. Specifically, in the Discussion section:

We tried to find some pattern in the PTs choice between competing instruments in each domain, but it seems (in line with Haigh et al’s., 2001 study (29)) that the use of outcome measures is  related to the specific regional contexts rather than to the psychometric properties or appropriateness of the instrument. One apparent pattern is in the less frequent use of specific outcome measures, such as those related to walking, pain/fatigability – energy level and major life areas,  in the Eastern region.

MS is an individually variable and unpredictable disease, which needs evaluation at different assessment levels (impairment, disability, handicap, quality of life), including patient-reported outcome measures. Our team, the Special Interest Group in Mobility of RIMS, has a long-term interest in understanding what is the most valuable assessment approach to address all levels of patient/therapist perspectives. Firstly, in the narrative review, drawbacks in current practice and research were defined - one of them being the non-systematic use of many different measures, making it difficult to directly compare the effectiveness of interventions. Secondly, several teams of specialists have tried to develop a core set of clinical functional assessments for MS rehabilitation [(31, 32)], others are working on gaining consensus to harmonize both assessments and interventions [(33)], but still without clear recommendations. Resources, such as the Rehabilitation Measures database (https://www.sralab.org/rehabilitation-measures), which details operational instructions and the psychometric properties of many outcome measures, is helpful for facilitating the use of outcome measures in clinical practice and designing  research. Thirdly, a prospective multicentre trial was realized with the aim of identifying the most relevant measurement of mobility for use in physical rehabilitation of pwMS (34). Responsiveness, clinically meaningful improvement, and real changes of measures demonstrate great heterogenity. For the mildly disabled pwMS the Multiple Sclerosis Walking Scale–12 and the physical subscale of the Multiple Sclerosis Impact Scale–29 were the most sensitive measures in detecting improvements in functional mobility after physical rehabilitation, while for the moderate to severely disabled pwMS, the Rivermead Mobility Index and the capacity test 5-Repetition Sit-to-Stand Test seemed to be useful (34).

Two decades ago, disability and functional independence were the predominant domains measured in neurological patients [(30)]. Our survey indicates that nowadays, a broad range of body function and activity domain measures are utilized by most (80%) of the respondents. This may be reflecting a shift in clinical practice towards a more systematic use of the ICF model. This model offers very good tools for assessment (ICF Assessment Sheet), goal setting (ICF Categorical Profile), for documentation of interdisciplinary treatment process (ICF Intervention Table) and for evaluation as to whether goals are addressed (ICF Evaluation Display), see (https://www.icf-casestudies.org). We are convinced that this person-centred concept has significant  potential for the future, because it reflects the components of individual functioning together with the contextual components (10).

Introduction

In the introduction section, the reader may feel a little bit unsatisfied about the characteristics and importance of the quality of physiotherapy and related activities in the context of multiple sclerosis itself. The problem should be formulated more precisely.

In the introduction, the paragraph characterising physical therapy in MS and emphasising the importance of the quality of physiotherapy was added including relevant references. Specifically:

Physical therapy (PT) primarily treats physical functions with the aim of promoting functional independence, preventing complications, and enhancing the overall quality of life. To achieve this, PT uses a variety of techniques and methods that can be broadly divided into four categories: physical activity (fitness/endurance/resistance) training, motor/skill acquisition (individualized therapy led intervention), neuroproprioceptive “facilitation, inhibition”, and technology-based PT [(4).  PT interventions can lead to improvements on all categories defined by the International Classification of Functioning, Disability and Health (ICF) (5, 6), which has been documented by clinically meaningful patient-reported outcomes (7-9).

Materials

  • line 108, bracket error

The error was corrected.

Result

The basic and most important problem that the Authors of this article must solve is the very poor quality of presented graphics. Almost all of the presented figures are completely unreadable. 

The values in the charts are illegible, which was already a big problem in the reviewing of this work. I am asking for the Authors to print a paper version of the paper and try to guess without a magnifying glass what is presented on e.g Fig 4 (this seems to be one of the most interesting results of this work, but unfortunately invisible). The reviewer does not know if this is a problem at the editorial level of the article, but necessarily it must be solved.

Thank you for this important point:

In order to make the figures readable, we have split original Figure 1A+B into two separate figures (1 and 2), and original Figure 2A+B into separate figures (3 and 4).

For original Figure 3 (now Figure 5) we have prepared Figures with better resolution (larger font), but these need to be combined into new Figure 5 (we are still working on this).  

For original Figure 4, we think we again need to split this into two separate tables (new Figure 6 and 7, see attached). However, I would need you to cut the white space around Figure 6, add the boxes with description of the clusters (we are still working on this).
Finally, we are still working on new Supplementary Table additional to Figure 3 with named the Domains. 

We are sorry, but we are still working on improvement of the graphs – it is technically more demanding than we expected. May we kindly ask you to postpone the term for graphs preparation?

Discussion

In the section results (lines 178-189) Authors mention several different indexes or scale types. In Discussion, the Authors characterize what kind of scale is using more frequently. Which of them in the Authors opinion bring the most valuable information? I understand that the purpose of the work was the- characterization, but the thematic scope of discussion could be enriched by the attempt of assessment of physical tests quality in the context of MS

Opinions have now been incorporated  into the discussion, including relevant references. Specifically, please see our answer above.

Other comments:

The statistical analysis in the Reviewer's opinion is performed properly, but there is a lack of justification for using the parametric tests?

We believe the use of the t test is well justified even for Likert-scale data, also see  de Winter, J. F.C. and Dodou, D. (2010) "Five-Point Likert Items: t test versus Mann-Whitney-Wilcoxon (Addendum added October 2012)," Practical Assessment, Research, and Evaluation: Vol. 15 , Article 11. DOI: https://doi.org/10.7275/bj1p-ts64

Reviewer 2 Report

This manuscript on goal setting and ways to measure outcome of physical therapies is very informative and well written. Would it be possible to color differently the diverse goal and assessment proportions in figures 1 and 2?

I have no recommendations for the authors to improve the content of their manuscript but for the suggestion of coloring differently the 5-7 items in figures 1 and 2 instead of using shadows of blue to differentiate them only. I ticket the box “….minor revision” because of this only. Really minor! Being the journal digital the colors will not produce additional costs and it will be easier for the readers to identify differences.

Author Response

Prague, June 22, 2020

Dear Editors,

Thank you very much for the review of our manuscript entitled ““Real-world goal setting and use of outcome measures according to the International Classification of Functioning Disability and Health (ICF): A European survey of physical therapy practice in multiple sclerosis” by Řasová K., Martinková P., Soler B., Freeman J., Cattaneo D., JonsdottirJ., Smedal T., Romberg A., Henze T., Santoyo-Medina C. and Feys P.

We have revised the manuscript according to reviewers’ suggestions. Changes are highlighted throughout the text of the paper. Responses to each reviewer’s comment are also commented separately below.

We are glad that the reviewers’ view the paper as raising important questions, informative and well written. We believe that our revisions have further improved the manuscript and hope that it is now suitable for publication  in the International Journal of Environmental Research and Public Health.

Sincerely,

                                                           Assoc. prof. PhDr. Kamila Řasová, Ph.D.This manuscript on goal setting and ways to measure outcome of physical therapies is very informative and well written.

Thank you for your positive feedback.

Would it be possible to color differently the diverse goal and assessment proportions in figures 1 and 2?

Thank you for the notice that graphs were not well readable. They were purposely prepared in different shades of blue color, similar to our previous related work (Martinkova et al., 2018).

To improve quality, in order to make the figures readable, we have split original Figure 1A+B into two separate figures (1 and 2), and original Figure 2A+B into separate figures (3 and 4).

For original Figure 3 (now Figure 5) we have prepared Figures with better resolution (larger font), but these need to be combined into new Figure 5 (we are still working on this).  

For original Figure 4, we think we again need to split this into two separate tables (new Figure 6 and 7, see attached). However, I would need you to cut the white space around Figure 6, add the boxes with description of the clusters (we are still working on this).
Finally, we are still working on new Supplementary Table additional to Figure 3 with named the Domains. 

We are sorry, but we are still working on improvement of the graphs – it is technically more demanding than we expected. May we kindly ask you to postpone the term for graphs preparation?

I have no recommendations for the authors to improve the content of their manuscript but for the suggestion of coloring differently the 5-7 items in figures 1 and 2 instead of using shadows of blue to differentiate them only. I ticket the box “….minor revision” because of this only. Really minor! Being the journal digital the colors will not produce additional costs and it will be easier for the readers to identify differences.

Round 2

Reviewer 1 Report

Dear Authors 

Thank you for all the corrections made. I believe that now the article meets all  standards of the journal and recommend it for publication in "International Journal of Environmental Research and Public Health"